# Investigation of UCHL3 and HNMT Gene Polymorphisms in Greek Patients with Type 2 Diabetes Mellitus and Diabetic Retinopathy

**DOI:** 10.3390/biomedicines13020341

**Published:** 2025-02-03

**Authors:** Konstantinos Flindris, Vivian Lagkada, Aikaterini Christodoulou, Maria Gazouli, Marilita Moschos, Georgios Markozannes, George Kitsos

**Affiliations:** 1Department of Ophthalmology, University General Hospital of Ioannina, 45500 Ioannina, Greece; xristodoulouk@yahoo.gr (A.C.); gkitsos@uoi.gr (G.K.); 2Laboratory of Biology, Department of Basic Medical Sciences, Medical School, National and Kapodistrian University of Athens, 11527 Athens, Greece; vivian.lagada@gmail.com (V.L.); mgazouli@med.uoa.gr (M.G.); 31st Department of Ophthalmology, University of Athens, 11527 Athens, Greece; moschosmarilita@yahoo.fr; 4Department of Hygiene and Epidemiology, Medical School, University of Ioannina, 45500 Ioannina, Greece; g.markozannes@uoi.gr

**Keywords:** diabetic retinopathy, diabetes mellitus, UCHL3, rs4885322, HNMT, rs11558538

## Abstract

Background and Objectives: Recent studies have shed light on the association between genetic factors and diabetic retinopathy (DR) onset and progression. The purpose of our study was to investigate the association between rs4885322 single-nucleotide polymorphism (SNP) of the *UCHL3* gene and rs11558538 SNP of the *HNMT* gene with the risk of DR in Greek patients with type 2 diabetes mellitus (T2DM). Materials and Methods: In our case–control study, we included 85 T2DM patients with DR and 71 T2DM patients without DR (NDR), matched by ethnicity and gender. Demographic and clinical data of all patients were collected, and then patients went through a complete ophthalmological examination and were genotyped for rs4885322 SNP of *UCHL3* gene and for the rs11558538 SNP of *HNMT* gene. Statistical analysis was implemented by STATA v.16.1. Results: No significant differences in demographic and clinical data were observed between the DR and the NDR group (*p*-value ≥ 0.05), except for the lower mean of age, longer duration of DM, more frequent use of insulin therapy, and higher levels of hemoglobin A1c (HbA1c) in the DR group. The allelic effect of rs488532 increases the risk of DR by 2.04 times, and in the dominant genetic model, the risk of DR is elevated by 123%, while both associations are statistically significant (*p*-value < 0.05). Moreover, the allelic effect of rs11558538 is associated with a 3.27 times increased DR risk and, in the dominant genetic model, reveals an augmented risk of DR by 231%, while both associations are also statistically significant (*p*-value < 0.05). Conclusions: The rs4885322 SNP of the *UCHL3* gene and the rs11558538 SNP of the *HNMT* gene are associated with DR risk in Greek patients with T2DM. However, further studies with larger samples and different ethnicities should be implemented to clarify the exact association of these SNPs and DR onset.

## 1. Introduction

Diabetic retinopathy (DR) is a microvascular and neurodegenerative disease of the retina and is considered the most common and serious ocular complication of diabetes mellitus (DM) [1]. DR is the leading cause of preventable blindness in working-age people (20–74 years old) in developed countries [2]. Globally, 246 million people are diagnosed with DM, and the prevalence of DM is expected to reach 629 million by 2045 [3]. Approximately one-third of people with DM develop signs of DR, and one-third or these people have vision-threatening complications [4]. Moreover, 75% of patients with type 1 DM (T1DM) and 50% of patients with type 2 DM (T2DM) will develop DR sometime in their life [5].

DR is classified based on the Early Treatment Diabetic Retinopathy Study (ETDRS) into non-proliferative DR (NPDR) with microaneurysms, hard exudates, cotton-wool spots, intraretinal hemorrhage, venous beadings and intraretinal microvascular abnormalities (IRMA), and proliferative DR (PDR) with external neovascularization and preretinal or vitreous hemorrhage [6]. Diabetic Macular Edema (DME) refers to thickening of the macula and may occur in either NPDR or PDR, and along with PDR, it composes the common vision-threatening complications of DR [7].

Diabetic microangiopathy through hyperglycemia and chronic inflammation leads to capillary occlusion and thereby to retinal ischemia, elevated vascular permeability, leakage, and neovascularization involving multiple biochemical signal pathways [8]. There are many factors leading to DR, such as the duration of DM, glycemic control, hypertension, dyslipidemia, nephropathy, stroke, smoking, higher Body Mass Index (BMI), pregnancy, anemia, and cataract surgery; however, the exact pathogenesis remains still elusive [9]. Despite the above risk factors, studies have revealed a substantial variation in the onset and progression of DR among different populations, and the complex etiology of DR reflects the sundry treatments currently available, including anti-VEGF and glucocorticoid intravitreal medications, laser photocoagulation, and vitrectomy [10]. Although the etiopathogenesis of DR has been studied thoroughly, the precise underlying mechanisms have not been clarified, and genetic factors have shown to play a pivotal role in DR development and severity [11].

*UCHL3* (ubiquitin carboxyl-terminal hydrolase L3) belongs to deubiquitinating enzymes that are capable of removing ubiquitin (Ub) from protein substrates and generate free monomeric Ub, thus maintaining the dynamic balance of intracellular Ub levels [12]. The conjunction of Ub and Ub-like proteins to intracellular proteins has emerged as a critical regulatory cellular process of paramount importance for numerous signaling pathways [13]. The *UCHL3* gene is located on chromosome 13q22.2 and has been associated with oncogenesis, for instance, with breast cancer and cervical carcinoma [14]. Furthermore, *UCHL3* has deneddylating activity essential for various cellular functions and promotes insulin signaling and adipogenesis, hence contributing to retinal maintenance in stress conditions [15]. One study has investigated the association between *UCHL3* single-nucleotide polymorphisms (SNPs), such as rs4885322 (A>G), and risk of DR, but the results were inconclusive [16].

The *HNMT* (histamine N-methyltransferase) gene produces an enzyme that catalyzes the N-methylation of histamine, which is a major metabolic pathway of histamine, responsible for the termination of its neurotransmitter action [17]. This gene is located on chromosome 2q22.1 and consists of six exons with a length of approximately 34 kilobases [18]. The rs11558538 (C314T) SNP is located in exon 4, and it is related to decreased enzyme activity and is associated with reduced risk of Parkinson’s disease [19]. *HNMT* takes part in the insulin signaling pathway; insulin activates insulin receptor (INSR) tyrosine kinase, which phosphorylates various proteins and activates different signaling pathways; and in the retina, disrupted insulin receptor (INSR) signaling leads to cellular disfunction [11].

Based on these observations, we decided to investigate the association between rs4885322 SNP of the *UCHL3* gene and rs11558538 SNP of the *HNMT* gene with the risk of DR, making this paper the first in the literature to conduct such genotyping in Greek patients with DM.

## 2. Materials and Methods

### 2.1. Study Design and Population

In our case–control study, we included 85 T2DM patients with DR and 71 T2DM patients without DR (NDR), matched by ethnicity and gender, who were recruited from the Department of Ophthalmology of the University General Hospital of Ioannina from January 2023 to June 2023. This study was conducted in accordance with the Declaration of Helsinki and was approved by the Institutional Review Board of University General Hospital of Ioannina (Protocol Code, 17835; and Date of Approval. 20 July 2022). In addition, written informed consent was obtained from all the participants before entering the study, and all samples were anonymized.

DM was defined according to the American Diabetes Association diagnostic criteria [20]. Patients with T1DM were excluded from the study, and there were no other exclusion criteria. Demographic and clinical data, including gender, age, onset and duration of DM, insulin or other antidiabetic medication used, BMI, smoking (packs/year), alcohol consumption, hemoglobin A1c levels (HbA1c), presence of hypertension, dyslipidemia, cardiovascular disease, stroke, anemia, pregnancy, nephropathy, cataract surgery, and other medical or ophthalmological commodities. Hypertension was defined in all patients who had antihypertensive treatment or systolic blood pressure (SBP) ≥ 140 mmHg or diastolic blood pressure (DBP) ≥ 90 mmHg. Dyslipidemia was determined if the patient had antilipemic agents or total cholesterol levels ≥ 200 mg/dL. Last but not least, nephropathy was identified when glomerular filtration rate (GFR) levels were <60 mL/min/1.73 m^2^, calculated with MDRD formula [21].

All the patients went through a complete ophthalmological examination, including the assessment of the best-corrected visual acuity using the Standard Snellen Chart, slit lamp biomicroscopy, tonometry using Goldman applanation, and fundoscopy after pupillary dilation. In cases where needed, fluorescein angiography was performed to confirm the diagnosis. Optical Coherence Tomography (OCT) and OCT–Angiography (OCT-A) were performed in all patients to determine the presence of DME. DR was diagnosed based on the ETDRS diagnostic criteria.

### 2.2. Genotyping

Samples of peripheral blood (approximately 2 mL) were collected from all patients. Genomic DNA was isolated from peripheral blood samples using the NucleoSpin Blood Kit (MACHEREY-NAGEL GmbH & Co. KG, Düren, Germany), according to the manufacturer’s instructions. The quality and concentration of purified DNA were estimated using the NanoDrop 8000 Spectrophotometer (Thermo Fisher Scientific Inc., Waltham, MA, USA). Regarding the rs11558538 polymorphism, the genotyping was performed as described by Dai et al. [22]. Briefly, polymerase chain reaction–restriction fragment length polymorphism (PCR-RFLP) was conducted to identify SNPs. The PCR system consisted of 5 μL of 2× Master Mix (Kapa Biosystems, Wilmington, MA, USA), 10 ng of genomic DNA, 0.2 μM of forward primer, and 0.2 μM of reverse primer (Eurofins Genomics AT GmbH, Vienna, Austria) [22]. For rs11558538, PCR amplification was performed with an initial denaturation step at 94 °C for 5 min, then 30 cycles of denaturation at 94 °C for 30 s, followed by annealing at 62 °C for 30 s, extension at 72 °C for 30 s, and eventually elongation at 72 °C for 4 min. The PCR products were digested with *Eco*R V at 37 °C for 1 h in 10 μL of H buffer containing 2 μL of PCR product, 1 μL of 10× H buffer, 6.75 μL of double-distilled H_2_O, and 0.25 units of restriction enzymes (TaKaRa Biotechnology, Shiga, Japan). The enzyme-digested products were isolated on 2% agarose gel, and the fragments were visualized on UV light [22]. Regarding the rs4885322 polymorphism, genotyping was performed by allele-specific PCR. The sequences of the primers were as follows: common reverse, 5′ TTTCTAATACTTTCAATCCACA 3; forward A allele, 5′ TATTGAGTTAGTGAGTAGAAAATAA 3′; and forward G allele, 5′ TATTGAGTTAGTGAGTAGAAAATAG 3′.

### 2.3. Statistical Analysis

The baseline characteristics of the study population were reported using descriptive statistics. Continuous variables were reported as medians with interquartile ranges (IQRs), and categorical variables as frequencies and percentages.

A case–control analysis was conducted to investigate the association between genetic variants and diabetic DR. We used logistic regression models to assess the association of SNPs and DR, adjusting for sex and age. Results are reported as odds ratios (ORs), along with 95% confidence intervals (CIs).

All SNPs were analyzed using three genetic models: (1) additive (risk increases per each additional risk allele, e.g., for rs4885322: AA = 0, AG or GA = 1, GG = 2), (2) dominant (one or two copies of the risk allele have the same effect on the outcome, e.g., AG/GG vs. AA), and (3) recessive (both risk alleles are required to affect the outcome, e.g., GG vs. AG/AA).

The association between genetic variants and the type of DR (proliferative vs. non-proliferative) was examined among participants with DR. Logistic regression models were employed using similar genetic models (additive, dominant, and recessive), as described above.

We further performed a sensitivity analysis after excluding participants with extreme BMI values (<18.5 or >40 kg/m^2^) to assess the robustness of the results.

Finally, to explore the impact of genetic variants on the duration of diabetic retinopathy, cumulative hazard curves were generated. We performed log-rank tests to compare curves between genotypes.

All statistical analyses were performed using Stata v. 16.1. Results were considered statistically significant at a *p*-value < 0.05.

## 3. Results

The descriptive characteristics of the NDR and DR patients are presented in Table 1. No significant differences were observed between the two groups in regard to gender status, BMI, smoking, cataract surgery, presence of glaucoma, hypertension, dyslipidemia, cardiovascular disease, stroke, anemia, nephropathy, hypothyroidism, and benign prostate hyperplasia (*p*-value ≥ 0.05). Although the DR group showed a lower mean of age, patients presented a significantly longer duration of DM and were more often under insulin therapy (*p*-value < 0.05). Additionally, patients with DR had elevated levels of HbA1c, and almost three out of four patients (74.1%) of this group had NPDR.

The genotyping results are also introduced in Table 1, and all of them were in the Hardy–Weinberg equilibrium. Among our patients, the frequency of the rs4885322 heterozygous genotype and the frequency of the rs11558538 heterozygous genotype were elevated in the DR group, in contrast with the NDR group. However, the presence of the rs4885322 homozygous genotype and the presence of the rs11558538 homozygous genotype in both groups were scarce.

Table 2 demonstrates the genetic analyses of the rs4885322 *UCHL3* gene SNP and of the rs11558538 *HNMT* gene SNP in the DR and the NDR groups based on sundry genetic models. First of all, concerning the rs4885322 SNP, the allelic effect increases the risk of DR, and the association is statistically significant (*p*-value = 0.04). Additionally, in the AG+GG vs. AA model the risk of DR is elevated, and the result is also statistically significant (*p*-value = 0.03). These associations remain statistically significant, when extreme BMI are excluded (*p*-value < 0.05).

The effect of the presence of the rs11558538 alleles on the development of DR is associated with increased risk, and the result is statistically significant (*p*-value = 0.01). Moreover, the CT+TT vs. CC model reveals an augmented risk of DR, while the allelic contrast highlights an elevated risk of DR, and both of these associations are statistically significant (*p*-value < 0.05). In fact, these correlations remain also statistically significant when extreme BMI values are excluded (*p*-value < 0.05).

Furthermore, the association of the rs4885322 and the rs11558538 SNPs with the development of PDR was examined in various genetic models (Table 3). Nevertheless, no statistically significant associations were identified between the rs4885322 and the rs11558538 SNPs and the onset of PDR in any genetic model (*p*-value > 0.05).

In Figure 1 and Figure 2, the cumulative hazard estimate of DR in association with the duration of DM was investigated according to rs11558538 and rs4885322 SNPs, whereas no statistically significant associations were identified between the hazard estimate of DR and NDR according to the rs4885322 and the rs11558538 SNPs (*p*-value > 0.05).

## 4. Discussion

Although DR is the most prevalent vascular disease of the retina and the leading ocular complication of DM [23], the precise etiopathogenetic mechanisms remain incompletely understood [24]. Current therapeutic approaches are insufficient to prevent or mitigate the complications associated with DR [25]. Major risk factors for DR include the duration of DM and poor glycemic control, and several other risk factors have been identified, including the presence of hypertension, dyslipidemia, obesity, cardiovascular disease, stroke, nephropathy, smoking, pregnancy, anemia, and prior cataract surgery [26]. The prevalence of DR also varies significantly across different ethnic groups. Notably, variations in DR risk and the complexity of the disease may be explained by genetic factors, such as gene mutations and abnormal gene expression, which contribute substantially to both the onset and progression of DR [27]. Genetic predispositions have been implicated in the development of DR through multiple biological signaling pathways, including insulin signaling, angiogenesis, neurogenesis, inflammation, adipogenesis, and the interaction between endothelial cells and leukocytes [28,29].

In this study, we sought to investigate the potential association between two SNPs, rs4885322 in the UCHL3 gene and rs11558538 in the HNMT gene, and the onset of DR in a Greek population. Our findings suggest that the rs4885322 SNP is significantly correlated with an increased risk of DR, a result consistent with a previous study in a Chinese population [16]; however, that study did not find statistically significant results. This is, to our knowledge, the first report in the literature to examine this SNP in relation to DR in a Caucasian population, providing new evidence of its potential role in elevating DR risk in T2DM patients. Despite this association, the precise molecular mechanisms underlying this relationship remain elusive. It has been proposed that the rs4885322 SNP may influence UCHL3 gene expression, disrupting cellular Ub levels, which in turn could impair insulin signaling, promoting insulin resistance and hyperglycemia [11].

In addition, the rs11558538 gene polymorphism is associated with the development of DR in our findings. This is also the first time this SNP has been examined in connection with DR in a Caucasian population. This SNP is thought to reduce the enzymatic activity of HNMT, leading to elevated histamine levels. This dysregulation may interfere with INSR signaling, contributing to chronic hyperglycemia [11]. These results further support the hypothesis that genetic factors play a critical role in the etiopathogenesis of DR and provide new insights into the potential molecular mechanisms involved.

Despite these important findings, some limitations should be acknowledged. The relatively small sample size and the focus on a Caucasian population may limit the generalizability of our results. Over and above that, the low number of patients with PDR precludes a meaningful analysis of the association between these two polymorphisms and the progression of DR to more severe stages. Moreover, the potential differences in DM duration, insulin therapy regimens and HbA1c levels between NDR and DR patients were not fully accounted for. Given our primary focus on genetic determinants of DR, our findings provide valuable insights into the genetic architecture of DR etiopathogenesis, laying groundwork for future studies. Despite these limitations, the current study elucidates the association between the rs4885322 UCHL3 gene SNP and the rs11558538 HNMT gene SNP and the risk of DR in Greek patients with T2DM.

## 5. Conclusions

To summarize, the role of the rs4885322 UCHL3 gene SNP and of the rs11558538 HNMT gene SNP was investigated in correlation with the risk of DR, and both are associated with DR risk in Greek patients with T2DM. Considering the complexity of the disease and the interaction between genetic and environmental factors, further studies with larger samples and different ethnicities should be implemented to clarify the exact association between these SNPs and DR risk.

## Figures and Tables

**Figure 1 biomedicines-13-00341-f001:**
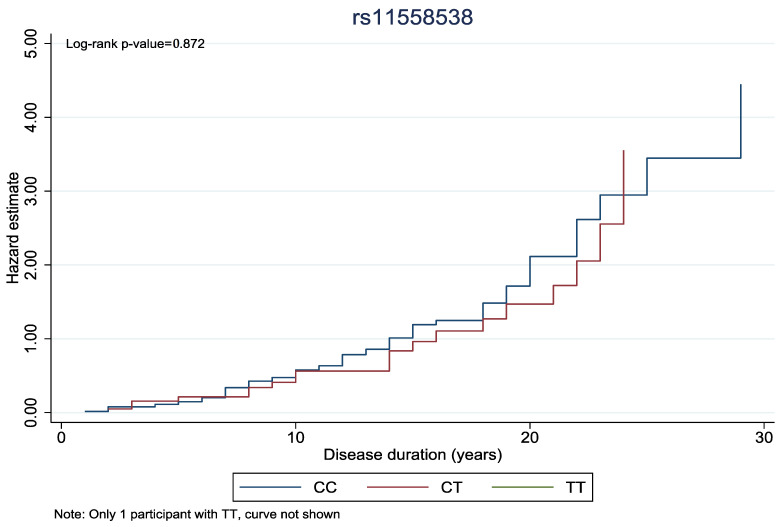
Cumulative hazard estimate of DR in association with the duration of DM according to rs11558538 SNP, presenting no statistically significant association between the CC homozygotes and CT heterozygotes.

**Figure 2 biomedicines-13-00341-f002:**
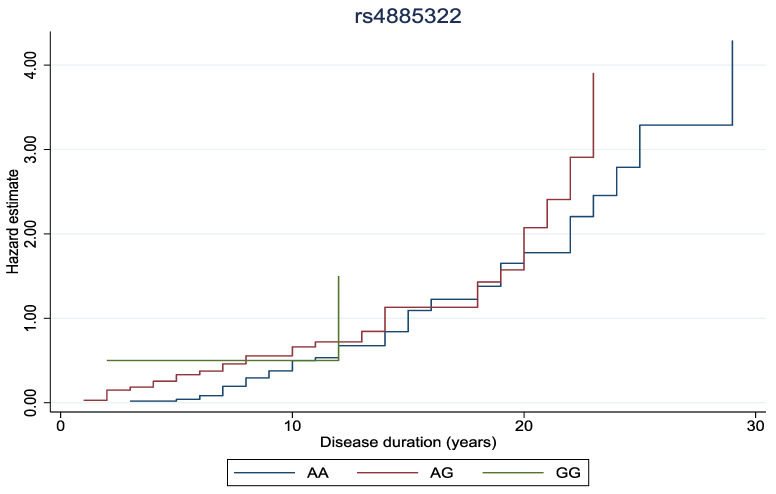
Cumulative hazard estimate of DR in association with the duration of DM according to rs4885322 SNP, illustrating no statistically significant association between the CC homozygotes, CT heterozygotes, and TT homozygotes.

**Table 1 biomedicines-13-00341-t001:** Descriptive characteristics. Results presented as median [interquartile range] or *n* (%).

Variable	NDR *n* = 71 (45.5)	DR *n* = 85 (54.5)	Total *n* = 156 (100)	*p*-Value
Female	28 (39.4)	26 (30.6)	54 (34.6)	0.25
Age (years)	74 [67–80]	69 [65–74]	72 [65.3–77]	0.01
BMI (kg/m^2^)	27.7 [25.7–30.4]	27.8 [25.9–30.3]	27.7 [25.7–30.3]	0.95
DM duration (years)	14 [6–18]	21 [13.5–29]	17 [10–26.75]	<0.01
HbA1c (%)	6.5 [6.1–7]	6.9 [6.3–7.2]	6.7 [6.2–7.1]	0.01
NPDR	-	63 (74.1)	-	
Duration DR (years)	-	12 [7–18]	-	
Macular Diabetic Edema	-	61 (71.8)	-	
Argon Laser Photocoagulation	0 (0)	19 (22.4)	19 (12.2)	<0.01
Intravitreal injections	9 (12.7)	65 (76.5)	74 (47.4)	<0.01
Vitrectomy	8 (11.3)	19 (22.4)	27 (17.3)	0.07
Cataract surgery	29 (40.8)	40 (47.1)	69 (44.2)	0.44
Glaucoma	11 (15.5)	19 (22.4)	30 (19.2)	0.28
Dry AMD	23 (32.4)	5 (5.9)	28 (17.9)	<0.01
Wet AMD	17 (23.9)	0 (0)	17 (10.9)	<0.01
Tablets	70 (98.6)	65 (76.5)	135 (86.5)	<0.01
Insulin	7 (9.9)	35 (41.2)	42 (26.9)	<0.01
Hypertension	57 (80.3)	67 (78.8)	124 (79.5)	0.82
Smoker	47 (66.2)	56 (65.9)	103 (66)	0.97
Smoking (pack-years)	31 [0–65]	41 [0–64.5]	36 [0–65]	0.59
Alcohol	14 (19.7)	14 (16.5)	28 (17.9)	0.60
Cardiovascular disease	27 (38)	40 (47.1)	67 (42.9)	0.26
Stroke	4 (5.6)	6 (7.1)	10 (6.4)	0.72
Dyslipidemia	49 (69)	70 (82.4)	119 (76.3)	0.05
Anemia	0 (0)	1 (1.2)	1 (0.6)	0.36
Nephropathy	4 (5.6)	7 (8.2)	11 (7.1)	0.53
Ca	7 (9.9)	1 (1.2)	8 (5.1)	0.01
Benign prostate hyperplasia	5 (7)	6 (7.1)	11 (7.1)	1
Hypothyroidism	7 (9.9)	5 (5.9)	12 (7.7)	0.35
rs4885322				
AA	55 (77.5)	49 (57.6)	104 (66.7)	
AG	15 (21.1)	34 (40)	49 (31.4)	
GG	1 (1.4)	2 (2.4)	3 (1.9)	0.03
rs11558538				
CC	64 (90.1)	64 (75.3)	128 (82.1)	
CT	7 (9.9)	20 (23.5)	27 (17.3)	
TT	0 (0)	1 (1.2)	1 (0.6)	0.05

Abbreviations: NDR, non-diabetic retinopathy; DR, diabetic retinopathy; BMI, Body Mass Index; DM, diabetes mellitus; HbA1c, hemoglobin A1c; NPDR, Non-Proliferative Diabetic Retinopathy; AMD, age-related macular degeneration.

**Table 2 biomedicines-13-00341-t002:** Genetic analyses of rs4885322 and rs11558538 SNPs on the development of DR.

	Main		Excluding Extreme BMI (*n* = 3)	
	OR (95% CI)	*p*-Value	OR (95% CI)	*p*-Value
rs4885322				
per allele	2.04 (1.03, 4.04)	0.04	2.02 (1.01, 4.04)	0.05
GG vs. AA+AG	1.35 (0.10, 18.00)	0.82	1.31 (0.10, 16.97)	0.84
AG+GG vs. AA	2.23 (1.07, 4.64)	0.03	2.22 (1.06, 4.67)	0.04
G vs. A	1.87 (0.98, 3.55)	0.06	1.86 (0.97, 3.58)	0.06
rs11558538				
per allele	3.27 (1.27, 8.39)	0.01	3.59 (1.34, 9.62)	0.01
TT vs. CC+CT	-	-	-	-
CT+TT vs. CC	3.31 (1.27, 8.61)	0.01	3.64 (1.34, 9.89)	0.01
T vs. C	3.05 (1.24, 7.52)	0.02	3.36 (1.30, 8.67)	0.01

Abbreviations: BMI, Body Mass Index; OR, odds ratio; CI, confidence interval.

**Table 3 biomedicines-13-00341-t003:** Genetic analyses of the rs4885322 and the rs11558538 SNPs on the development of PDR.

	RR (95% CI)	*p*-Value
rs4885322		
per allele	0.53 (0.19, 1.48)	0.225
AG+GG vs. AA	0.55 (0.18, 1.64)	0.283
G vs. A	0.60 (0.24, 1.51)	0.279
rs11558538		
per allele	0.56 (0.17, 1.86)	0.345
CT+TT vs. CC	0.57 (0.16, 2.00)	0.382
T vs. C	0.58 (0.18, 1.85)	0.360

Abbreviations: RR, risk ratio; CI, confidence interval.

## Data Availability

Data available upon request.

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
