# Peer review of "Investigation of UCHL3 and HNMT Gene Polymorphisms in Greek Patients with Type 2 Diabetes Mellitus and Diabetic Retinopathy"

_biomedicines, 2025, doi:10.3390/biomedicines13020341_

Round 1
Reviewer 1 Report
Comments and Suggestions for Authors
The paper investigates the association of UCHL3 (rs4885322) and HNMT (rs11558538) gene polymorphisms with diabetic retinopathy (DR) in Greek patients with type 2 diabetes mellitus (T2DM). Statistical analysis revealed significant associations between both single nucleotide polymorphisms (SNPs) and DR risk, although the results are limited by the small sample size. Overall, the paper is well-prepared.
1. Please provide reference for this sentence: “We demonstrated that the rs4885322 SNP is correlated with the risk of developing DR and these findings are in accordance with a Chinese study, that presented similar findings, which although were not statistically significant.”
2. (Sharma et al., 2019) was mentioned a few times in the discussion. Please use the numbered format which aligns with other references in this paper.
Author Response
Dear Reviewer,
Thank you for your thorough review and valuable feedback on our manuscript. We appreciate the time and effort you have dedicated to improving the quality of our work. We have carefully considered your comments and suggestions, and have revised the manuscript accordingly, as it follows.
Comment 1: [Please provide reference for this sentence: “We demonstrated that the rs4885322 SNP is correlated with the risk of developing DR and these findings are in accordance with a Chinese study, that presented similar findings, which although were not statistically significant.”]
Response 1: [ Agree and we have revised the manuscript accordingly]
Thank you for pointing this out.
Comment 2: [ (Sharma et al., 2019) was mentioned a few times in the discussion. Please use the numbered format which aligns with other references in this paper.]
Response 2: [Agree and we have revised the manuscript accordingly]
Thank you for pointing this out.
Reviewer 2 Report
Comments and Suggestions for Authors
The manuscript entitled “The Role of UCHL3 and HNMT Gene Polymorphisms in Greek Patients with Diabetic Retinopathy” focuses on the genetic basis underlying the individual predisposition for diabetic retinopathy (DR) in patients with T2DM. The topic of the manuscript is within the scope of the journal, would be interesting to the readership, especially to the clinicians and researchers interested in molecular basis of T2DM and clinical management of diabetic states, provided that the study design is adequate for reaching the intended study aim.
The manuscript is relatively well written and well structured. Material and methods presented most of the required details, while the presentation of the Results is informative and the major conclusions are supported by the findings. There are still some issues that should be resolved and some corrections are needed:
- “The role” in the title of the manuscript should be avoided, since it implies that a mechanistic study was conducted. Furthermore, it should be highlighted in the title that the study refers to T2DM patients.
- Line 133: what were primer sequences? If primers from previous study by Dai et al. were used, this should be specified.
- Line 139: There is a missing information for the amount of restriction enzyme (0.25 of what? Units? microliters? If microliters, the concentration should be specified). When performing allele-specific PCR, an adequate method for eliminating the possibility of mistyping would be to include an internal control in a multiplex PCR reaction. This control is lacking in the present study.
- All results should be adjusted for the duration of disease, which is a major confounding factor. Segregation of DR patients according to the type is rational, but not with this small number of participants, which can be a source of bias.
- Differences between patients with DR and controls in terms of DM duration, frequency of insulin therapy and HbA1c levels are major issues regarding the results, as all of these parameters are major confounders. This should be highlighted and discussed with a critical appraisal of the findings.
- OR is wrongfully interpreted. The interpretation of this value as an increase in risk is mathematically more complex and what the authors stated as “2.04 times”, “123%” and “3.27 times” is appropriate for relative risk, not for odds ratio. Therefore, the descriptive interpretation of the results presented in tables in incorrect. It should be stated that the presence of certain allele increases/decreases risk, without the exact number, or with a statement saying that OR value is exactly some specific number.
- Hazard estimate is unreliable, because of a small sample size and the allele frequency.
Author Response
Dear Reviewer,
Thank you for your thorough review and valuable feedback on our manuscript. We appreciate the time and effort you have dedicated to improving the quality of our work. We have carefully considered your comments and suggestions, and have revised the manuscript accordingly, as it follows.
Comment 1: [“The role” in the title of the manuscript should be avoided, since it implies that a mechanistic study was conducted. Furthermore, it should be highlighted in the title that the study refers to T2DM patients.]
Response 1: [The new title is: "Investigation of UCHL3 and HNMT Gene Polymorphisms in Greek Patients with Type 2 Diabetes Mellitus and Diabetic Retinopathy"]
Thank you for pointing this out
Comment 2: [Line 133: what were primer sequences? If primers from previous study by Dai et al. were used, this should be specified.]
Response 2: [The primers were used from previous study by Dai et al. and it is corrected in the manuscript]
Thank you for pointing this out.
Comment 3: [Line 139: There is a missing information for the amount of restriction enzyme (0.25 of what? Units? microliters? If microliters, the concentration should be specified). When performing allele-specific PCR, an adequate method for eliminating the possibility of mistyping would be to include an internal control in a multiplex PCR reaction. This control is lacking in the present study.]
Response 3: [The amount of 0.25 Units of restriction enzymes was used]
Thank you again for this.
Comment 4: [All results should be adjusted for the duration of disease, which is a major confounding factor. Segregation of DR patients according to the type is rational, but not with this small number of participants, which can be a source of bias.]
Response 4: [ We definitely agree and we have revised the manuscript accordingly in lines 269 - 272. "Moreover, the potential differences in DM duration, insulin therapy regimens and HbA1c levels between NDR and DR patients were not fully accounted for. Given our primary focus on genetic determinants of DR, our findings provide valuable insights into the genetic architecture of DR etiopathogenesis, laying groundwork for future studies." ]
Thank you for pointing this out.
Comment 5: [Differences between patients with DR and controls in terms of DM duration, frequency of insulin therapy and HbA1c levels are major issues regarding the results, as all of these parameters are major confounders. This should be highlighted and discussed with a critical appraisal of the findings.]
Response 5: [ We definitely agree and we have revised the manuscript accordingly in lines 269 - 272. "Moreover, the potential differences in DM duration, insulin therapy regimens and HbA1c levels between NDR and DR patients were not fully accounted for. Given our primary focus on genetic determinants of DR, our findings provide valuable insights into the genetic architecture of DR etiopathogenesis, laying groundwork for future studies." ]
Thank you for pointing this out.
Comment 6: [OR is wrongfully interpreted. The interpretation of this value as an increase in risk is mathematically more complex and what the authors stated as “2.04 times”, “123%” and “3.27 times” is appropriate for relative risk, not for odds ratio. Therefore, the descriptive interpretation of the results presented in tables in incorrect. It should be stated that the presence of certain allele increases/decreases risk, without the exact number, or with a statement saying that OR value is exactly some specific number.]
Response 6: [Agree and we have revised the manuscript accordingly.]
Thank you for pointing this out.
Comment 7: [Hazard estimate is unreliable, because of a small sample size and the allele frequency.]
Response 7: [We acknowledge that the relatively modest sample size and allele frequency in our study may impact the precision of our hazard estimates. We recognize this limitation and emphasize that these preliminary results should be interpreted cautiously and warrant validation in larger cohorts.]
Thank you once again for your guidance and support.
Round 2
Reviewer 2 Report
Comments and Suggestions for Authors
The authors have replied to my comments and significantly improved the manuscript.
Author Response
Thank you for your timely review of our revised manuscript. We are pleased to hear that our revisions have addressed your concerns and significantly improved the paper. Your thoughtful feedback throughout this process has been invaluable in strengthening our work.